# BRCA2 C-Terminal RAD51-Binding Domain Confers Resistance to DNA-Damaging Agents

**DOI:** 10.3390/ijms23074060

**Published:** 2022-04-06

**Authors:** Zida Zhu, Taisuke Kitano, Masami Morimatsu, Arisa Tanaka, Ryo Morioka, Xianghui Lin, Koichi Orino, Yasunaga Yoshikawa

**Affiliations:** 1Laboratory of Veterinary Biochemistry, School of Veterinary Medicine, Kitasato University, Aomori 034–8628, Japan; zzd01@icloud.com (Z.Z.); kitano@vmas.kitasato-u.ac.jp (T.K.); arisa@sunchase.jp (A.T.); vm12135q@st.kitasato-u.ac.jp (R.M.); vm17098@st.kitasato-u.ac.jp (X.L.); orino@vmas.kitasato-u.ac.jp (K.O.); 2Laboratory of Laboratory Animal Science and Medicine, Department of Disease Control, Graduate School of Veterinary Medicine, Hokkaido University, Sapporo 060–0818, Japan; mmorimat@vetmed.hokudai.ac.jp

**Keywords:** BRCA2, DNA damage repair, homologous recombination repair, C-terminal RAD51-binding domain, X-ray irradiation, mitomycin C

## Abstract

Breast cancer type 2 susceptibility (BRCA2) protein is crucial for initiating DNA damage repair after chemotherapy with DNA interstrand crosslinking agents or X-ray irradiation, which induces DNA double-strand breaks. BRCA2 contains a C-terminal RAD51-binding domain (CTRBD) that interacts with RAD51 oligomer-containing nucleofilaments. In this study, we investigated CTRBD expression in cells exposed to X-ray irradiation and mitomycin C treatment. Surprisingly, BRCA2 CTRBD expression in HeLa cells increased their resistance to X-ray irradiation and mitomycin C. Under endogenous BRCA2 depletion using shRNA, the sensitivities of the BRCA2-depleted cells with and without the CTRBD did not significantly differ. Thus, the resistance to X-ray irradiation conferred by an exogenous CTRBD required endogenous BRCA2 expression. BRCA2 CTRBD-expressing cells demonstrated effective RAD51 foci formation and increased homologous recombination efficiency, but not nonhomologous end-joining efficiency. To the best of our knowledge, our study is the first to report the ability of the BRCA2 functional domain to confer resistance to X-ray irradiation and mitomycin C treatment by increased homologous recombination efficiency. Thus, this peptide may be useful for protecting cells against X-ray irradiation or chemotherapeutic agents.

## 1. Introduction

Tumor therapy includes X-ray irradiation and chemotherapy, which induce double-strand breaks (DSBs) in DNA [1,2]. Although these treatments are directed toward tumor cells, they often have harmful side effects on normal cells. Improved tumor therapies aim to increase tumor cell death and reduce normal cell death [3,4]. Therefore, radiation and chemotherapy sensitizers and protectors have been developed to modulate these undesirable side effects. Preventing or enhancing DNA repair pathways is a popular attribute used for developing sensitizers or protectors, respectively [5]. DSBs are repaired by two major DNA repair pathways, namely, homologous recombination (HR) repair and nonhomologous end joining (NHEJ), which maintain the integrity of genomic DNA [6]. While NHEJ is error-prone, HR repair is a high-fidelity, template-dependent mechanism because it uses the intact genome information from sister chromatids and functions during the S and G2 phases. Moreover, HR can contribute to DNA interstrand crosslinks (ICLs), which can be induced by certain antitumor agents, such as cisplatin or mitomycin C (MMC) [7]. Therefore, with chemicals previously developed to decrease or increase DNA repair efficiency, controlling HR repair efficiency has potential for applications in developing chemotherapy sensitizers or protectors [8,9].

The DNA recombinase RAD51 and its regulator BRCA2 participate in HR repair [10]. When the function of either protein is impaired, cells become highly susceptible to DSB- and ICL-inducing treatments [11,12,13]. To regulate the HR repair activity of RAD51, BRCA2 carries two RAD51-binding domains: one central domain featuring eight BRC repeats and a C-terminal RAD51-binding domain (CTRBD) containing a phosphorylation site [14,15] (Figure 1A). Although RAD51 binding varies among these eight BRC repeats, the interaction between any BRC repeat and a RAD51 monomer facilitates the loading of RAD51 onto single-stranded DNA, thereby forming a nucleofilament—a complex composed of a RAD51 oligomer and single-stranded DNA—representing the first step in HR repair [15]. The CTRBD functions in HR repair by stabilizing RAD51 oligomers and nucleofilaments [16,17]. However, this contribution is not as large because the effect of the mutant BRCA2 with non-RAD51-interacting mutation in the CTRBD manifests when it is assayed with a peptide-fused BRCA2 essential domain, but not with the full-length BRCA2 protein [18]. Further, the RAD51 binding of this domain is controlled by phosphorylation at S3291 and T3387. Unlike a dephosphorylated domain at these two sites, the phosphorylated BRCA2 C-terminus at these two sites does not interact with RAD51 [15,19].

Ovarian cancers harboring BRCA2 mutations have increased sensitivity against chemotherapy [20]. This report indicates the insufficiency of the BRCA2 function to confer sensitivity to chemotherapy and probably to X-ray irradiation. Furthermore, the expression of the BRC repeat domain inhibits RAD51 nucleofilament formation in vivo and in vitro; thus, this domain is considered a promising drug target candidate based on its strong interaction with the RAD51 monomer and ability to inhibit oligomerization [21,22,23]. However, the potential therapeutic application of the CTRBD remains unknown.

We hypothesized that the expression of an exogenous CTRBD, which interacts with a RAD51 oligomer, would effectively interfere with HR repair. We first assessed peptides around the BRCA2 C-terminus that strongly interacted with RAD51. To evaluate the therapeutic application of the CTRBD, we tested the sensitivity of cells, particularly X-ray irradiation- and MMC-treated cells expressing the BRCA2 CTRBD, to DNA-damaging treatments and then functionally analyzed HR.

## 2. Results

### 2.1. Phosphorylation Sites and the C-Terminal Flanking Region of CTRBD Affect RAD51 Interaction

The interaction of the CTRBD with RAD51 is regulated by the phosphorylation status at S3291 and T3387 [15,19]. However, there are currently no integrated reports on the mechanism of interaction. To identify candidate sensitizer peptides, we reassessed the mutations that influenced RAD51 binding. We tested the RAD51 interaction of the BRCA2 C-terminal region (3260–3418 aa) with the CTRBD using coimmunoprecipitation. The BRCA2 C-terminal region interacted with RAD51, and this interaction was enhanced by T3387D or T3387E phosphorylation-mimicking mutants. However, this finding was not observed in the T3387A mutation, which disrupted phosphorylation (Figure 1B). Then, we examined the effect of the C-terminal flanking region of the CTRBD. The immunoprecipitated peptides of the shortened CTRBD had a lower yield than those of the C-terminal region, but the yields of the coimmunoprecipitated RAD51 were comparable (Figure 1C). Thus, the RAD51 interaction of the shortened peptide was stronger than that of the whole C-terminus. Using the mammalian two-hybrid assay, we then assessed three mutants of the CTRBD in which S3291 was substituted by cysteine (S3291C) or glycine (S3291G) or was deleted (delS3291) to prevent phosphorylation at this site (Figure 1D). The P3292L mutation was used as the positive control because it abolishes the phosphorylation motif of cyclin-dependent kinase (CDK) and the constitutive interaction with RAD51 [24]. RAD51 interaction was abolished in all three S3291 mutants; similarly, the phosphorylation-mimicking S3291E mutant did not interact with RAD51. The P3292L mutant interacted with RAD51, but its interacting activity was slightly weaker than that of the wild type, as previously reported [24]. We also performed a coimmunoprecipitation assay to investigate the characteristics of the P3292L mutant (Figure 1E). The expression of the P3292L mutant was lower than that of the wild type or the S3291E mutant; however, its interaction with RAD51 persisted.

### 2.2. CTRBD but Not BRCA2 C-Terminus Confers Resistance to X-ray Irradiation and MMC Treatment

We then tested the effects of the exogenously expressed CTRBD on the sensitivity of cells to X-ray irradiation and MMC treatment by using HeLa cells that stably expressed the FLAG-HA-fused CTRBD. The expression of the CTRBD containing the P3292L mutation was lower than that of the wild type and the S3291E mutant (Figure 2A). The expression of the CTRBD containing the S3291E mutation was higher than that of the wild type. The resistance of wild-type and P3292L-mutated CTRBD-expressing cells to X-ray irradiation and MMC treatment significantly increased compared with that of the empty vector-transduced HeLa cells, but this increased resistance was not observed in the S3291E mutation-expressing cells (Figure 2B,C). This resistance to X-ray irradiation was also exhibited by HepG2 cells (Appendix A).

We also tested the sensitivity of BRCA2 C-terminus-expressing HeLa cells to X-ray irradiation and MMC treatment (Figure 2D–F). Cell survival analysis showed that the sensitivity of the BRCA2 C-terminus- and FLAG-HA-tag-expressing cells to these treatments was comparable (Figure 2E,F).

### 2.3. Resistance to X-ray Irradiation Conferred by CTRBD Depends on Endogenous BRCA2

The BRCA2 CTRBD with the P3292L peptide might exhibit the strongest interaction among all peptides assessed. However, the expression of this mutant was too low and difficult to increase. Wild-type BRCA2 CTRBD-expressing cells exhibited a phenotype comparable with that of P3292L peptide-expressing cells. Thus, we used the wild-type peptide of the BRCA2 CTRBD for the subsequent experiments. We then investigated whether the CTRBD alone could confer resistance to DNA damage or whether endogenous BRCA2 was required. We also conducted the shRNA knockdown of BRCA2 in empty vector-transduced HeLa cells and HeLa cells expressing the wild-type or S3291E-mutated CTRBD (Figure 3A) and evaluated their sensitivity to X-ray irradiation or MMC treatment. The resistance of wild-type CTRBD-expressing HeLa cells to X-ray irradiation and MMC treatment significantly increased compared with that of scramble shRNA and empty vector-transduced HeLa cells, but that of the cells expressing the S3291E-mutated CTRBD did not (Figure 3B,C). Both BRCA2-knockdown strains—with or without the overexpression of the wild-type or S3291E-mutated CTRBD—exhibited a decreased resistance to X-ray irradiation and similar sensitivity levels (Figure 3B,C).

### 2.4. Exogenous Expression of BRCA2 CTRBD Affects HR Efficiency but Not NHEJ Efficiency

The exogenous expression of the BRCA2 CTRBD, together with endogenous BRCA2, decreased DNA damage sensitivity. We detected the HR and NHEJ efficiencies using two GFP-based assays [25,26,27]. We generated HeLa cells harboring a DR-GFP construct that expressed the wild-type BRCA2 or S3291E-mutated CTRBD, which lacked the capacity to interact with RAD51 (Figure 4A). The expression of the CTRBD containing the S3291E mutation was also higher than that of the wild type. We confirmed that the expression of the wild-type CTRBD protected the cells against MMC treatment (Figure 4B). Moreover, the HR repair efficiency increased in wild-type CTRBD-expressing cells but not in S3291E mutant-expressing or empty vector-transduced cells (Figure 4C).

We also assessed the NHEJ efficiency (Figure 5) and generated HeLa cells harboring an EJ5 construct that expressed the wild-type CTRBD, the S3291E-mutated BRCA2 CTRBD, or empty vector-transduced cells. We found that the expression of the S3291E-mutated BRCA2 CTRBD was higher than that of the wild-type CTRBD (Figure 5A). The NHEJ efficiency was similar in the empty vector and wild-type or S3291E mutant-expressing cells (Figure 5B).

To further clarify the mechanism underlying the increased efficiency of HR repair, we performed a RAD51 foci formation assay, which is commonly used to measure the recruitment efficiency of RAD51 to DNA damage sites. After X-ray irradiation, wild-type CTRBD-expressing cells generated RAD51 foci, similar to those in empty vector-transduced cells or CTRBD-expressing cells (Figure 6A). To further quantify the recruitment efficiency of RAD51 to DNA damage sites, we counted the number of RAD51 foci per cell in cyclin A-positive cells, which is the indicator of S to G2 phase progression (Figure 6B). Under normal conditions, the expression of the exogenous BRCA2 CTRBD had no significant effect on the RAD51 foci. In the presence of radiation-induced DNA damage, the BRCA2 CTRBD-expressing cells had more RAD51 foci than the empty vector or S3291E mutant-expressing cells.

### 2.5. CTRBD Expression Does Not Affect Cell Cycle Kinetics

HR efficiency increased possibly because of the accumulation of cells between the S and G2 phases due to the CTRBD overexpression. The effect of the exogenous CTRBD expression on the cell cycle was analyzed. The results indicate no major changes in the cell cycle distribution after the BRCA2 CTRBD overexpression (Figure 7).

## 3. Discussion

We hypothesized that the BRCA2 CTRBD, which interacts with the RAD51 oligomer and nucleofilaments, would interfere with HR repair-like peptides derived from BRC repeats. However, we found that the exogenous BRCA2 CTRBD expression conferred resistance to X-ray irradiation and MMC treatment in the presence of endogenous BRCA2. In one of the mechanisms involved, HR efficiency but not NHEJ efficiency was enhanced by BRCA2 CTRBD expression. The BRCA2 CTRBD interacts with the RAD51 oligomer or RAD51 nucleofilament, forming the HR initiation complex consisting of a single-stranded DNA wrapped around the RAD51 oligomer [16,17]. Although the CTRBD enhances RAD51 nucleofilament formation in vitro, the mechanism underlying the regulation of RAD51 nucleofilament formation and maintenance in vivo remains unclear. Wild-type CTRBD-transduced and empty vector-transduced cells effectively exhibited RAD51 foci formation. However, the number of RAD51 foci in wild-type CTRBD-expressing cells between the S and G2 phases, but not in the S3291E mutant, whose RAD51 interaction was abolished, increased after irradiation. Consistent with this result, HeLa and HepG2 cells expressing the S3291E mutant exhibited no resistance to MMC treatment. When the endogenous BRCA2 expression was downregulated, the sensitivity of empty vector-transduced and wild-type CTRBD-transduced HeLa cells to X-ray irradiation did not differ. Thus, the CTRBD alone was insufficient to recruit a critical number of RAD51 molecules to DSB sites and mediate DNA repair. These results suggest that the CTRBD expression supported the endogenous intact BRCA2 and improved the efficiency of RAD51 nucleofilament formation and HR repair after X-ray irradiation or MMC treatment; consequently, the cells became resistant to these treatments. Because HR was performed between the S and G2 phases, the HR efficiency increased possibly due to the altered distribution of cell cycle phases by the CTRBD overexpression. However, the cell cycle analysis showed that the CTRBD expression did not affect cell cycle kinetics.

We also determined the CTRBD expression level in HeLa cells. The comparison of the wild type and mutants (P3292L or S3291E) revealed that the CTRBD expression of the P3292L mutant, which induced a greater increase in the RAD51 interaction than the wild type, was lower than that of the wild type. Conversely, the CTRBD expression of the S3291E mutant without RAD51 interaction was higher than that of the wild type. These results show that the quantity and quality of remarkable RAD51 interactions were harmful to cells and that a moderate RAD51 interaction was required to resist DNA damage. This implication should be an important consideration in the application of this peptide to molecular drug development.

The full-length BRCA2 C-terminus (3260–3418 aa), which includes the CTRBD, had no effect on the sensitivity of the cells to X-ray irradiation and MMC treatment. The RAD51 interaction of the BRCA2 C-terminus was lower than that of the shortened peptide (3260–3331 aa). This result suggests that the extended region at the C-terminus, which possesses a phosphorylation site (T3387) activated by the checkpoint kinase Chk1 or Chk2 [19], did not directly interact with RAD51, but it reduced the affinity of BRCA2 to RAD51. In a previous study, the RAD51 interaction was decreased by phosphorylation or phosphorylation-mimicking mutations at T3387 [19]. However, unexpectedly, we detected an increased RAD51 interaction in the presence of the phosphorylation-mimicking mutations T3387D and T3387E, but not T3387A. The greatest difference between previous studies and ours lies in the transfected cell lines involved. In a previous study, the researchers used breast cancer-derived MCF7 cells; here, we used cervical cancer-derived HeLa cells [19]. The difference in the origin of cancer cells may have affected the role of phosphorylation at T3387 in the CTRBD. Although we did not further investigate this observation, we surmised that the process involving phosphorylation at T3387 was likely the mechanism through which the CTRBD was regulated by the C-terminal region of the domain.

The interaction between RAD51 and the CTRBD is regulated by the phosphorylation status at S3291 [15]. Thus, we assessed S3291 mutations that eliminated phosphorylation, resulting in a continuous interaction with RAD51. Previous reports indicated that glycine or alanine substitutions can abolish RAD51 interaction [15]. Here, we explored serine deletion and substitutions with cysteine, which is often used as a nonphosphorylated structural analog of serine. None of the mutations led to RAD51 interaction. We previously reported the core domain of the CTRBD [24] and showed that the N-terminal deletion including S3291 of the CTRBD interacting with RAD51 had a significantly lower intensity than the full length of the CTRBD. Hence, S3291 on the CTRBD is a crucial amino acid residue mediating the interaction between the CTRBD and RAD51.

In this study, we unexpectedly found that expressing the BRCA2 C-terminal binding domain in the presence of intact BRCA2 effectively recruited RAD51 to DNA damage sites, increased HR efficiency, and increased resistance to X-ray irradiation and MMC treatment. Although our work was limited to cell-based data, it provides substantial evidence supporting the potential of the BRCA2 CTRBD for applications in cell protection during radiation therapy and/or chemotherapy as a molecular drug that could enhance HR activities.

## 4. Materials and Methods

### 4.1. Cell Culture, Antibodies, and Generated Cell Lines

The HeLa, HepG2, and HEK293T cell lines were obtained from RIKEN Cell Bank and grown in Dulbecco’s modified Eagle’s medium supplemented with 10% fetal bovine serum. The following antibodies were used for Western blotting and immunostaining: anti-RAD51 (dilution 1:500–1000 H-92; Santa Cruz Biotechnology, Dallas, TX, USA), anti-BRCA2 (dilution 1:250 OP-95; Merck, Darmstadt, Germany), anti-lamin B1 (dilution 1:1000 PM064; MBL, Nagoya, Japan), anti-FLAG (dilution 1:1000 M2; Merck), anti-α-Tubulin (dilution 1:1000 M175-3; MBL), anti-GFP (dilution 1:1000 598; MBL), and anti-Cyclin A antibody (dilution 1:1000 B8; Santa Cruz Biotechnology).

The FLAG-HA- or FLAG-HA-EGFP-fused BRCA2 C-terminus (3260–3418 aa) and FLAG-HA-fused CTRBD (3260–3331 aa) were stably expressed using the pOZ-N plasmid [28]. The shRNA-mediated knockdown of BRCA2 was achieved by expressing the target sequence 5′-TACAATGTACACATGTAACAC-3′ in the pLKO.1 vector (donated by David Root; Addgene plasmid no. 10878) and scramble shRNA (donated by David Sabatini; Addgene plasmid no. 1864) [29,30]. The plasmid DNA was transfected into HEK293T cells using FuGENE HD (Promega, Madison, WI, USA) in accordance with the manufacturer’s instructions.

### 4.2. Immunoprecipitation

FLAG tag-fused peptides were subjected to immunoprecipitation [24]. In brief, HeLa cells that stably expressed the BRCA2 C-terminus or CTRBD were washed with phosphate-buffered saline (PBS), harvested by scraping, and centrifuged at 500× *g* for 5 min. The pelleted cells were resuspended with an equal volume of Benzonase buffer (20 mM Tris-HCl, pH 8.0; 50 mM KCl; 2 mM MgCl_2_; 0.5% Triton X-100; 10% glycerol; and 200 U/mL Benzonase) at 4 °C to digest the genomic DNA. The resuspended cells were subsequently treated with a 10 × IP buffer (20 mM Tris-HCl, pH 8.0; 150 mM KCl; 2 mM MgCl_2_; and 10% glycerol) to extract the proteins. The extracts were centrifuged at 17,000× *g* and 4 °C for 10 min, and the supernatants were combined with FLAG M2 affinity agarose (Merck) and incubated at 4 °C for 3 h. The immunoprecipitates were recovered by centrifugation at 500× *g* for 5 min and washed twice with the IP buffer. The samples were eluted into 50 μL of 1 × lithium dodecyl sulfate (LDS) loading buffer (Thermo Fisher Scientific, Waltham, MA, USA).

### 4.3. Mammalian Two-Hybrid Assay

Mammalian two-hybrid assays were performed as previously described [31]. Briefly, the coding regions of the BRCA2 CTRBD and RAD51 (GenBank ID: NM_000059 and NM_002875) were cloned into pM and pVP16 plasmids (Clontech, Mountain View, CA, USA), respectively. Approximately 2 × 10^5^ cells were seeded in 24-well plates and cotransfected with 50 ng of pVP, 50 ng of pM, 100 ng of pGluc, and 10 ng of pRL-TK (Promega). The cells were harvested 48 h after transfection, and luciferase activity was measured using the Dual-Luciferase Reporter Assay System (Promega). The transfection efficiency was normalized to the measured *Renilla* luciferase activity by using pRL-TK. The luciferase reporter plasmid pGLuc was generated as described previously [32]. At least three independent experiments were performed in triplicate for all experiments, and representative data are shown in each figure.

### 4.4. Clonogenic Assay

Cells were irradiated using an MX-80Labo (mediXtec, Chiba, Japan) at a dose of 1–6 Gy to evaluate the effects of X-ray irradiation. Then, approximately 100–4000 cells per well were seeded in six-well plates. For MMC treatment, approximately 200–400 cells per well were seeded in six-well plates and treated with 0.25–4 ng/mL MMC the following day. After 10–14 days, colony formation was assessed using 1% (*w*/*v*) crystal violet in methanol.

### 4.5. HR and NHEJ Assay

HeLa cells were transfected with the pDRGFP or pimEJ5GFP plasmid for assessing HR or NHEJ efficiency, respectively. The transfected cells were treated with 2 μg/mL puromycin and cloned in cell lines that became GFP-positive after the transfection of pCBASceI, the plasmid encoding I-SceI endonuclease. The pDRGFP, pCBASceI, and pimEJ5GFP plasmids were donated by Maria Jasin and Jeremy Stark (Addgene plasmid no. 26475, 26477, and 44026, respectively) [25,26,27].

The CTRBD or pOZ vector-transduced DR-GFP-harboring HeLa cells were generated using the method described above. These cells were seeded in six-well plates and cotransfected with 900 ng of pCBASceI and 100 ng of the DsRed monomer-encoding plasmid (Takara Bio, Shiga, Japan) for the HR assay. Under the same condition, the cells were cotransfected with 900 ng of pimEJ5GFP and 100 ng of the mCherry-encoding plasmid for the NHEJ assay. The same batch of plasmid mixture was used throughout. Four days after the transfection, the numbers of GFP-positive and DsRed- or mCherry-positive cells were determined via flow cytometry by using CytoFLEX (Beckman Coulter, Brea, CA, USA). DsRed or mCherry positivity rates were used to normalize transfection efficiency. For each sample, a minimum of approximately 20,000 cells were analyzed.

### 4.6. Immunostaining and Microscopy

HeLa cells were cultured on coverslips (Matsunami Glass, Osaka, Japan) and irradiated using an MX-80Labo (mediXtec) at a dose of 5 or 10 Gy. After 4, 6, and 24 h, the irradiated cells were harvested and fixed with 4% paraformaldehyde. After permeabilization with 0.3% Triton X-100 in PBS, the cells were incubated with anti-RAD51 antibody (dilution 1:500–1000 H-92; Santa Cruz Biotechnology). Anti-cyclin A antibody (dilution 1:1000 B8; Santa Cruz Biotechnology) was used to identify the S to G2 cell cycle phases. Alexa Fluor-568-conjugated goat anti-rabbit IgG (dilution 1:1000; Thermo Fisher Scientific) and Alexa Fluor 488 goat anti-mouse IgG (dilution 1:1000, Thermo Fisher Scientific) were used as secondary antibodies. Cell nuclei were stained with 4′,6-diamidino-2-phenylindole (DAPI; dilution 1:10,000, Thermo Fisher Scientific); RAD51 foci were examined under a fluorescence microscope (Leica Microsystems, Wetzlar, Germany). Cells with more than five RAD51 foci were considered RAD51 focus-positive cells. The RAD51 foci were examined under a confocal laser scanning microscope (Carl Zeiss AG, Oberkochen, Germany) with ZEISS ZEN Software (Carl Zeiss AG). The number of RAD51 foci present in the nuclei stained with DAPI and the signal of cyclin A were quantified.

### 4.7. Cell Cycle Analysis Using Flow Cytometry

The transduced cells were harvested using cold PBS, fixed in 40% ethanol at 4 °C, and stained with propidium iodide (PI) solution (PBS with 50 μg/mL PI, 100 μg/mL Rnase A, 0.1% Triton X-100, and 5% glycerol). The samples were analyzed on a CytoFLEX flow cytometer (Beckman Coulter Life Sciences, Indianapolis, IN, USA), and the cell cycle distribution was examined by FlowJo (Becton, Dickinson and Company, Franklin Lakes, NJ, USA).

### 4.8. Statistical Analysis

Mammalian two-hybrid, NHEJ, and RAD51 foci-counting assay results were statistically analyzed using one-way ANOVA followed by Dunnett’s test or Tukey’s HSD test in GraphPad Prism 8 (GraphPad Software, San Diego, CA, USA). Clonogenic, HR, and RAD51 foci formation assay results were statistically examined using an F-test followed by Student’s *t*-test with Holm’s correction in Microsoft Excel (Microsoft, Redmond, WA, USA). Statistical significance was set at *p* < 0.05 for all statistical analyses.

## Figures and Tables

**Figure 1 ijms-23-04060-f001:**
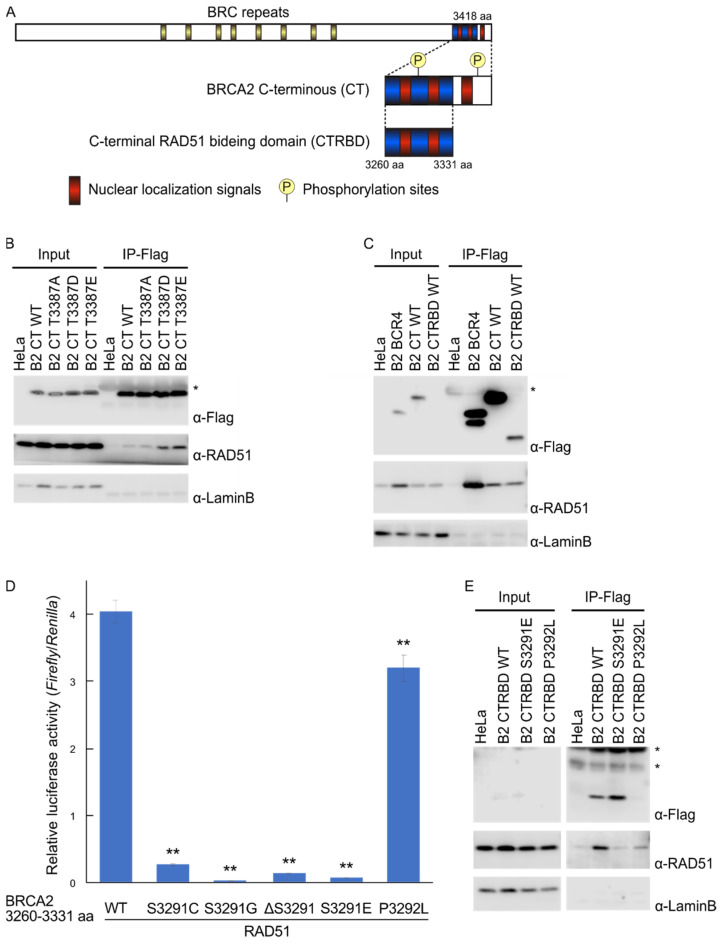
RAD51 interaction of the BRCA2 C-terminal RAD51-binding domain (CTRBD). (**A**) Schematic of the BRCA2 protein and its CTRBD. BRCA2 has two major sites to interact with RAD51: the BRC repeats and the C-terminal RAD51-binding region. Three nuclear localization signals and two phosphorylation sites (S3291 and T3387) are also indicated. (**B**) The interaction of RAD51 with the wild-type BRCA2 C-terminus (3260–3418 aa) and its phosphorylation-mimicking (T3387D and T3387E) and phosphorylation-eliminated (T3387A) mutants was assessed using coimmunoprecipitation. Lysates from these BRCA2 C-terminus-expressing HeLa cells were immunoprecipitated with anti-FLAG antibody. The immunoprecipitated samples were analyzed via Western blotting using anti-FLAG, anti-RAD51, and anti-laminB antibodies (negative control). Asterisk (*) indicate IgG light chain bands. (**C**) The RAD51 interactions of the BRCA2 C-terminus (3260–3418 aa) and CTRBD (3260–3331 aa) were compared by coimmunoprecipitation. BRC repeat 4 was used as the positive control. Lysates from BRCA2 C-terminus-expressing HeLa cells were immunoprecipitated with anti-FLAG antibody. The immunoprecipitated samples were analyzed via Western blotting using anti-FLAG, anti-RAD51, and anti-laminB antibodies (negative control). Asterisk (*) indicate IgG light chain bands. (**D**) Interactions between the BRCA2 CTRBD (3260–3331 aa) mutants (S3291C, S3291G, ΔS3291, S3291E, and P3292L) and RAD51 were evaluated using a mammalian two-hybrid assay. Lysate luciferase activity was determined 48 h after transfection. Results are presented as the mean, and error bars indicate the standard deviation (*n* = 3). Significance was examined by one-way ANOVA and Dunnett’s tests. Asterisks indicate significant differences between the wild-type CTRBD and the mutants (** *p* < 0.01). (**E**) The RAD51 interaction of the wild-type BRCA2 CTRBD (3260–3331 aa) and its phosphorylated mutants (S3291E and P3292L) was assessed by coimmunoprecipitation. Lysates from CTRBD-expressing HeLa cells were immunoprecipitated with anti-FLAG antibody. The immunoprecipitated samples were analyzed via Western blotting using anti-FLAG, anti-RAD51, and anti-laminB antibodies (negative control). Asterisks (*) indicate nonspecific bands.

**Figure 2 ijms-23-04060-f002:**
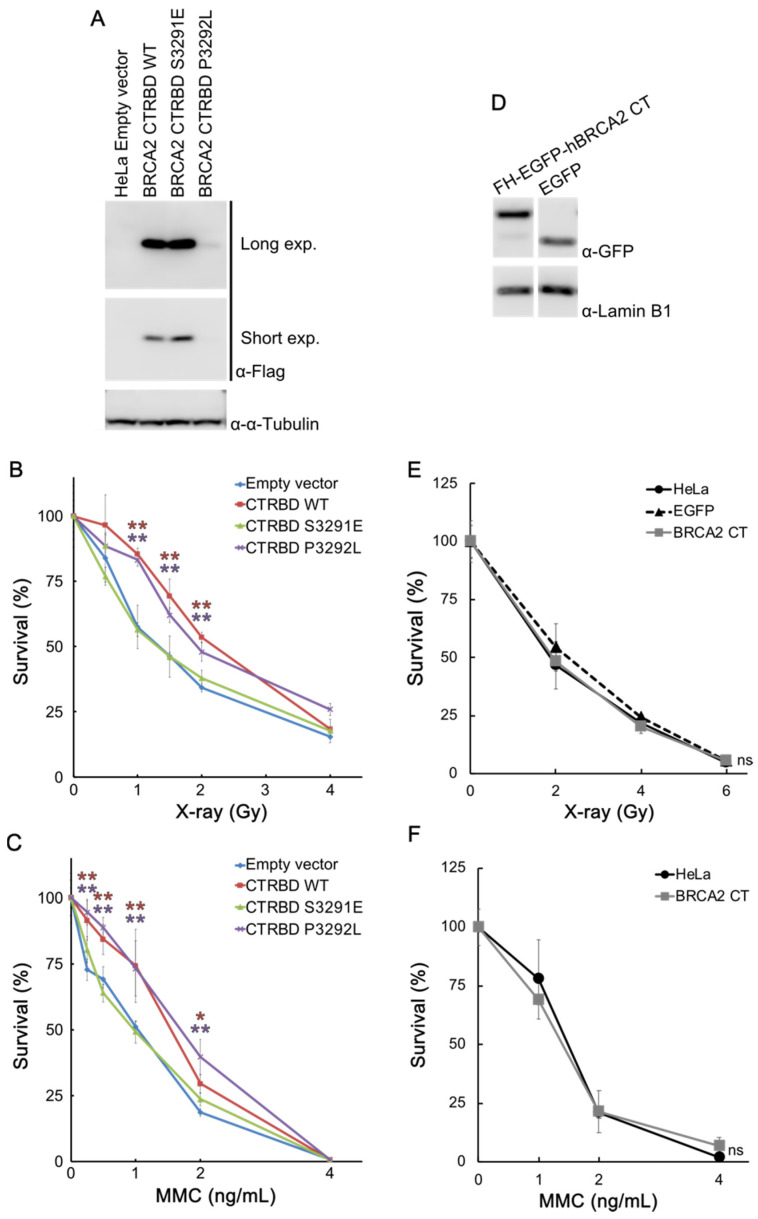
The BRCA2 CTRBD but not the BRCA2 C-terminus confers resistance to X-ray irradiation and mitomycin C (MMC) treatment. (**A**) The expression of the exogenous FLAG-HA-fused BRCA2 CTRBD (3260–3331 aa) and its mutants (S3291E and P3292L) was evaluated via Western blotting using anti-FLAG antibody. Long exp., long exposure; Short exp., short exposure. (**B**,**C**) A clonogenic survival assay of FLAG-HA-fused BRCA2 CTRBD (3260–3331 aa)-expressing cells treated with X-ray irradiation or MMC is shown. Results are presented as the mean at each X-ray or MMC dose, and error bars indicate the standard deviation (*n* = 3). Significance was examined by an F-test, followed by Student’s *t*-test with Holm’s correction. Asterisks indicate significant differences between empty vector-transduced HeLa cells and FLAG-HA-fused wild-type or P3292L-mutated BRCA2 CTRBD-expressing cells (* *p* < 0.05, ** *p* < 0.01). (**D**) Expression of the exogenous FLAG-HA-EGFP-fused BRCA2 C-terminus (3260–3418 aa) was determined by Western blotting using anti-GFP antibody. (**E**,**F**) A clonogenic survival assay of FLAG-HA-EGFP-fused BRCA2 C-terminus- and FLAG-HA-EGFP-expressing cells was performed after X-ray irradiation or MMC treatment. Results are presented as the mean at each X-ray or MMC dose, and error bars indicate the standard deviation (*n* = 3). Significance was examined by an F-test, followed by Student’s *t*-test with Holm’s correction. ns, non-significant; CT, C-terminus.

**Figure 3 ijms-23-04060-f003:**
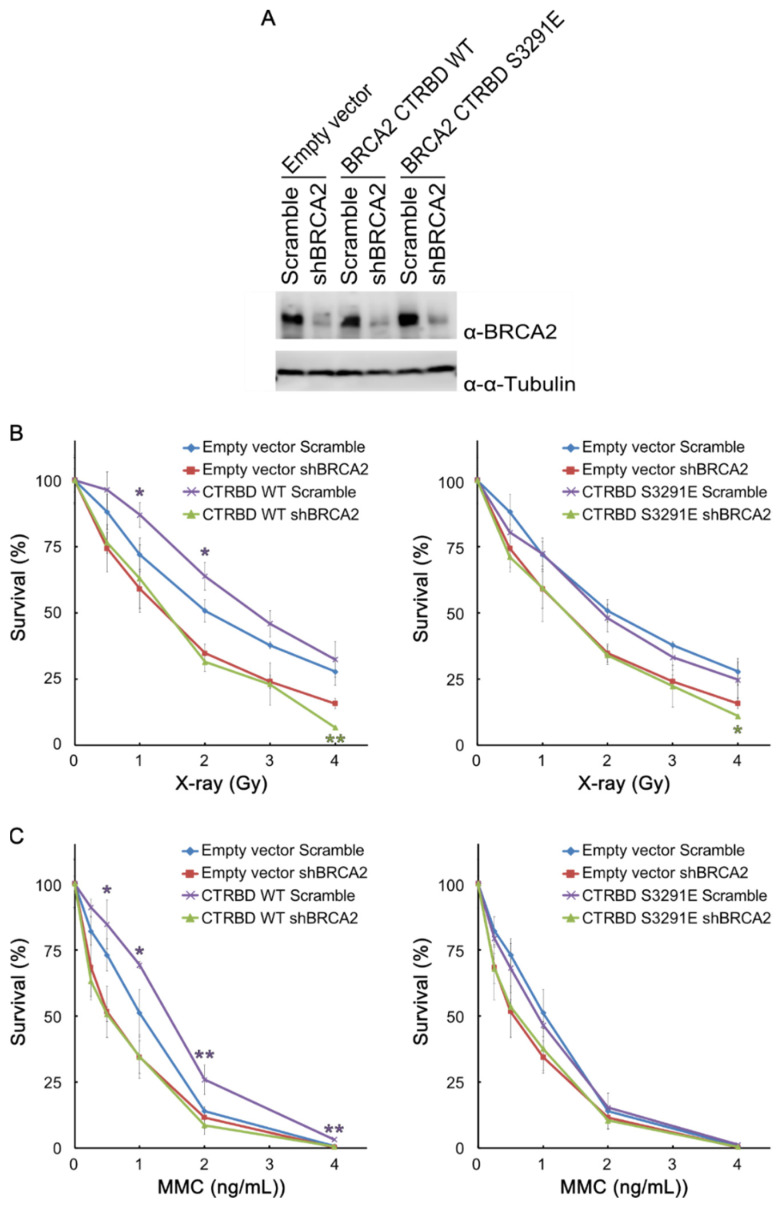
Resistance to X-ray irradiation conferred by the CTRBD expression depends on endogenous BRCA2. (**A**) Expression of endogenous BRCA2 in empty vector-transduced cells and wild-type or S3291E-mutated CTRBD-expressing cells was determined via Western blotting using anti-BRCA2 antibody in BRCA2-depleted HeLa cells generated by the shRNA-mediated knockdown of *BRCA2*. (**B**,**C**) A clonogenic survival assay of BRCA2-depleted cells after X-ray irradiation or MMC treatment is shown. Results are indicated as the mean at each X-ray or mitomycin C (MMC) dose, and error bars indicate the standard deviation (*n* = 3). Significance was examined by an F-test, followed by Student’s *t*-test with Holm’s correction. Asterisks indicate significant differences between empty vector-transduced HeLa cells and FLAG-HA-fused wild-type or S3291E-mutated BRCA2 CTRBD-expressing cells (* *p* < 0.05, ** *p* < 0.01).

**Figure 4 ijms-23-04060-f004:**
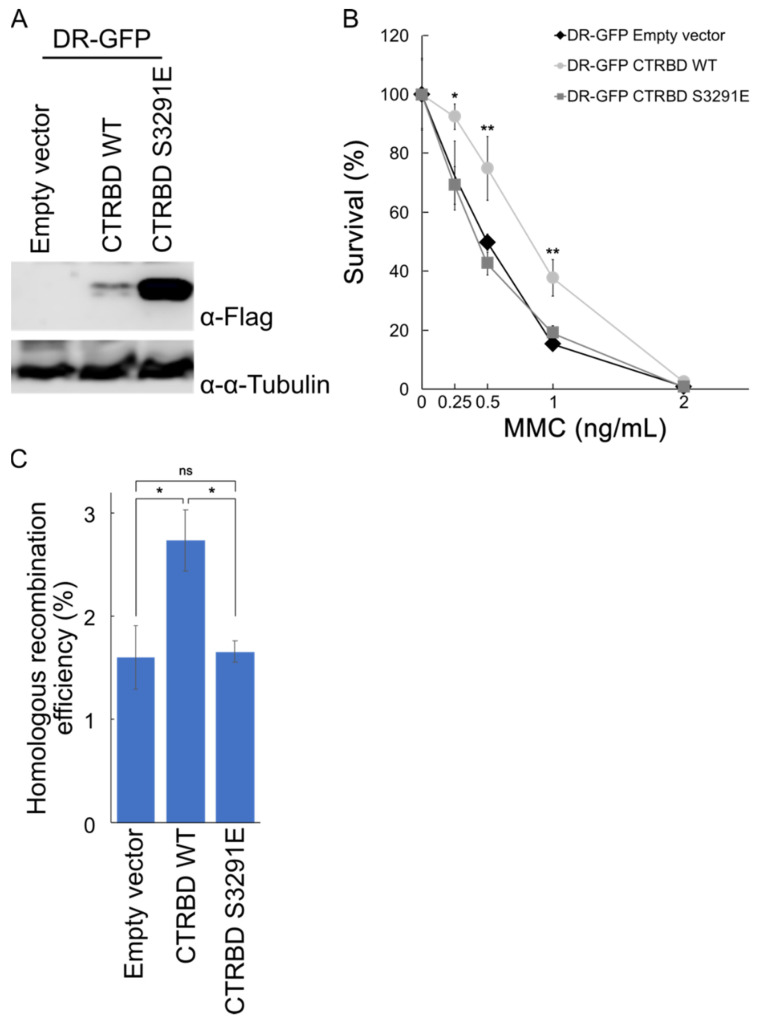
Expression of the CTRBD increases the homologous recombination repair efficiency. (**A**) Expression of the wild-type or S3291E-mutated CTRBD (3260–3331 aa) in HeLa cells harboring DR-GFP, a GFP-based homologous recombination efficiency-assessing construct, was determined via Western blotting by using anti-FLAG antibody. (**B**) A clonogenic survival assay of wild-type or S3291E-mutated FLAG-HA-fused BRCA2 CTRBD-expressing cells treated with mitomycin (**C**) (MMC) is shown. Results are presented as the mean at each MMC dose, and error bars indicate the standard deviation (*n* = 3). Significance was examined by an F-test, followed by Student’s *t*-test with Holm’s correction. Asterisks indicate significant differences between empty vector-transduced HeLa cells and S3291E mutant or wild-type BRCA2 CTRBD-expressing cells (* *p* < 0.05, ** *p* < 0.01). (**C**) Homologous recombination efficiency was tested using a GFP-based assay. The transfection efficiency was normalized by the cotransfection of the DsRed monomer-expressing vector. Significance was examined by an F-test, followed by Student’s *t*-test with Holm’s correction. Asterisks indicate significant differences (* *p* < 0.05, ** *p* < 0.01). The error bars indicate the standard deviation (*n* = 3). ns: non-significant.

**Figure 5 ijms-23-04060-f005:**
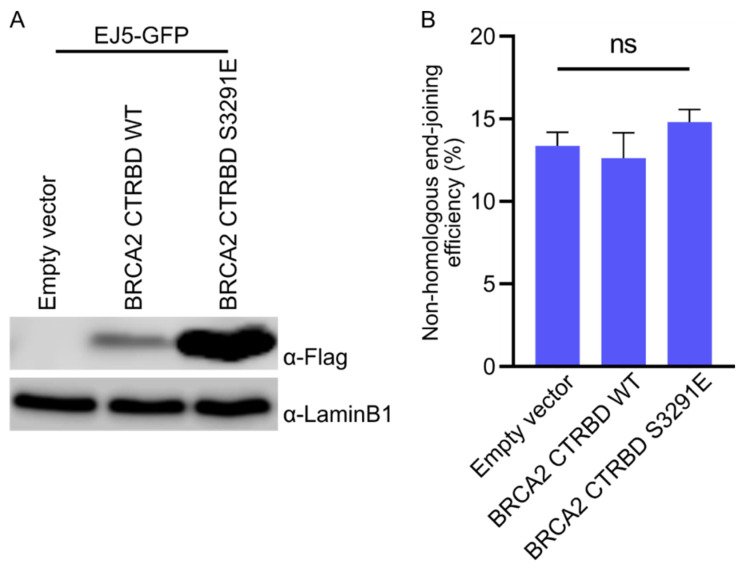
Expression of the CTRBD did not influence nonhomologous end-joining efficiency. (**A**) Expression of the exogenous FLAG-HA-fused BRCA2 CTRBD (3260–3331 aa) and its mutant (S3291E) was evaluated using Western blotting. (**B**) Nonhomologous end-joining efficiency was assessed using GFP-based assays. The transfection efficiency was normalized by the cotransfected mCherry-expressing vector. Significance was examined by one-way ANOVA with Tukey’s HSD multiple comparison test. The error bars indicate the standard deviation (*n* = 3). ns: non-significant difference in each control group.

**Figure 6 ijms-23-04060-f006:**
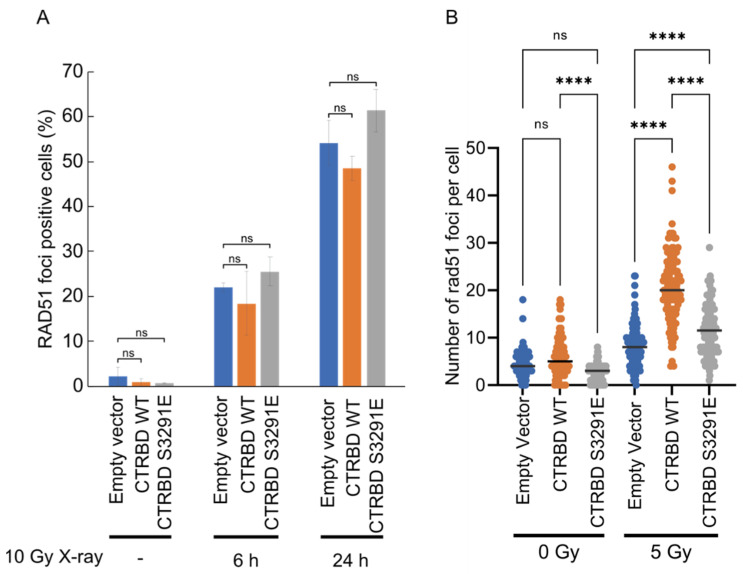
Exogenous expression of the BRCA2 CTRBD increases the number of RAD51 foci. (**A**) Wild-type or S3291E-mutated CTRBD (3260–3331 aa)-expressing cells were irradiated with X-ray at a dose of 10 Gy. These cells were harvested after 6 and 24 h. Cells with more than five RAD51 foci were considered RAD51 focus-positive cells. Results are presented as the mean, and error bars indicate the standard deviation (*n* = 3). Significance was examined using an F-test, followed by Student’s *t*-test with Holm’s correction. Results are indicated as the mean at each X-ray or MMC dose, and error bars indicate the standard deviation (*n* = 3). ns; non-significant. (**B**) Wild-type or S3291E-mutated CTRBD (3260–3331 aa)-expressing cells were irradiated with X-ray at a dose of 5 Gy and harvested after 4 h. The RAD51 foci per cell were counted and are presented as a dot plot. The lines indicate the median. Significance was examined using Student’s *t*-test with Holm’s correction or Tukey’s HSD multiple comparison test. Asterisks indicate significant differences (**** *p* < 0.0001).

**Figure 7 ijms-23-04060-f007:**
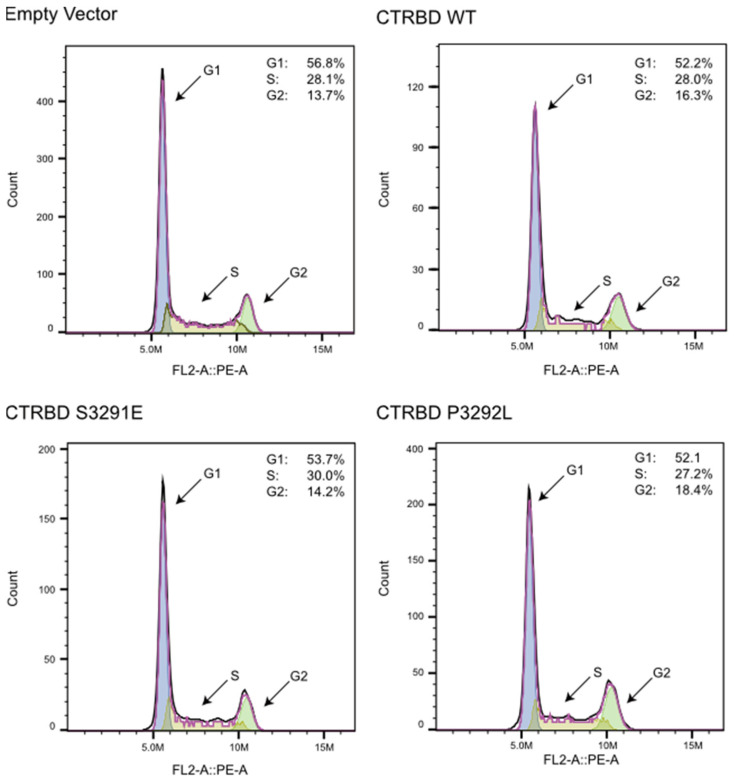
Cell cycle analysis of CTRBD-expressing cells. Flow cytometry histograms of cell populations of empty pOZ-N vector, wild-type, S3291E, and PS3292L mutant CTRBD (3260–3331 aa)-expressing cells in cell cycle phases. The percentage of each cell phase is indicated in the right upper space in each histogram.

## Data Availability

The datasets used and analyzed in the current study are available from the corresponding author upon reasonable request.

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
