# Peer review of "BRCA2 C-Terminal RAD51-Binding Domain Confers Resistance to DNA-Damaging Agents"

_ijms, 2022, doi:10.3390/ijms23074060_

Round 1
Reviewer 1 Report
The study is focused on the role of RAD51 and BCRA2 in DNA repair.
The hypothesis was that the expression of the exogenous CTRBD, which interacts with the AD51 oligomer, would effectively interfere with HR repair.
The authors found that the expression of the BRCA2 C-terminal binding domain in the presence of intact BRCA2 recruits RAD51 to DNA damage sites increases HR efficiency, and increases resistance to X-ray irradiation and MMC treatment. This work provides substantial evidence for the potential applications of the BRCA2 CTRBD in cell protection during radiation therapy and/or chemotherapy as a molecular drug that enhances HR activity.
The introduction is simple and clear, so it is the hypothesis. The experimental design is straightforward and results and data are well presented.
Minor revisions:
- Please, adjust the sentence in lines 55-57.
- Check the Fig 4B; is "DR-GFP CTRBD SE" correct? Or do you mean DR-GFP CTRBD S3291E?
- Each experiment is performed in triplicate (n=3), but how many independent experiments have been performed?
Reviewer 2 Report
The manuscript entitled “BRCA2C-terminal RAD51-binding domain confers resistance to DNA-damaging agents” focuses on the investigation of effects of expressing the CTRBD in cells exposed to X-ray irradiation and 15 mitomycin C treatment. Therefore, the authors were aimed to study molecular mechanisms underlying a role of DRCA2C in repair of DNA damage initiated by chemotherapy and X-ray irradiation; this makes the paper to be of interest to the journal audience. There are some concerns and recommendations:
- Introduction section: some sentences in paragraph 1, lines 29-35 (the first and the third ones) require references.
- Figure 1 should be moved to the Results section.
- Results, subsection 2.1, first sentence (line 105) and elsewhere: the authors stated that “The interaction of the CTRBD with RAD51 is known to be regulated by S3291 and T3387”. However, amino acid residues cannot regulate the interactions; instead, the interactions can be regulated by amino acid mutation or modifications such as phosphorylation, etc. Please, rephase.
- Results section, subsections 2.4. and 2.5 require more detailed analysis and interpretations of the results obtained. For example, Figs. 4C and 5A and 5B were not properly analyzed.
- References: citations of more recent papers in this field would be useful. For example: doi: 10.1016/j.yexcr.2021.112742; doi: 10.3390/genes12071034; doi: 10.1292/jvms.21-0006, etc.
